# Intestinal Anti-Inflammatory Effect of a Peptide Derived from Gastrointestinal Digestion of Buffalo (*Bubalus bubalis*) Mozzarella Cheese

**DOI:** 10.3390/nu11030610

**Published:** 2019-03-13

**Authors:** Gian Carlo Tenore, Ester Pagano, Stefania Lama, Daniela Vanacore, Salvatore Di Maro, Maria Maisto, Raffaele Capasso, Francesco Merlino, Francesca Borrelli, Paola Stiuso, Ettore Novellino

**Affiliations:** 1Department of Pharmacy, School of Medicine and Surgery, University of Naples Federico II, Via D. Montesano 49, 80131 Naples, Italy; giancarlo.tenore@unina.it (G.C.T.); ester.pagano@unina.it (E.P.); maria.maisto@unina.it (M.M.); francesco.merlino@unina.it (F.M.); ettore.novellino@unina.it (E.N.); 2Department of Precision Medicine, University of Campania “Luigi Vanvitelli”, Via De Crecchio 7, 80138 Naples, Italy; stefania.lama@unicampania.it (S.L.); daniela.vanacore@unicampania.it (D.V.); 3DiSTABiF, Università degli Studi della Campania “Luigi Vanvitelli”, via Vivaldi 43, 81100 Caserta, Italy; salvatore.dimaro@unicampania.it; 4Department of Agricultural Sciences, University of Naples Federico II, 80055 Portici, Italy; raffaele.capasso@unina.it

**Keywords:** intestinal inflammation, inflammatory bowel disease, epithelial adherens junctions, bioactive peptides

## Abstract

Under physiological conditions, the small intestine represents a barrier against harmful antigens and pathogens. Maintaining of the intestinal barrier depends largely on cell–cell interactions (adherent-junctions) and cell–matrix interactions (tight-junctions). Inflammatory bowel disease is characterized by chronic inflammation, which induces a destructuring of the architecture junctional epithelial proteins with consequent rupture of the intestinal barrier. Recently, a peptide identified by *Bubalus bubalis* milk-derived products (MBCP) has been able to reduce oxidative stress in intestinal epithelial cells and erythrocytes. Our aim was to evaluate the therapeutic potential of MBCP in inflammatory bowel disease (IBD). We studied the effect of MBCP on (i) inflamed human intestinal Caco2 cells and (ii) dinitrobenzene sulfonic acid (DNBS) mice model of colitis. We have shown that MBCP, at non-cytotoxic concentrations, both *in vitro* and *in vivo* induced the adherent epithelial junctions organization, modulated the nuclear factor (NF)-κB pathway and reduced the intestinal permeability. Furthermore, the MBCP reverted the atropine and tubocurarine injury effects on adherent-junctions. The data obtained showed that MBCP possesses anti-inflammatory effects both *in vitro* and *in vivo*. These results could have an important impact on the therapeutic potential of MBCP in helping to restore the intestinal epithelium integrity damaged by inflammation.

## 1. Introduction

Inflammatory bowel disease (IBD) is a term used to describe conditions that are characterized by chronic inflammation of the gastrointestinal tract. Crohn’s disease and ulcerative colitis are the principal types of inflammatory bowel disease. Worldwide, the prevalence and incidence of IBD is increasing mainly in newly industrialized countries whose societies have become more Westernized [1]. Although the etiology of IBD is still unknown, it arises as a result of the interaction of environmental or genetic factors and the immune responses [2]. IBD treatment involves the use of anti-inflammatory drugs (such as 5-aminosalicylic acid), corticosteroids, monoclonal antibodies [anti-tumor necrosis factor (TNF)-α antibodies] and vascular adhesion molecules [2,3] (anti-integrin antibodies). However, these drugs are not curative therapies and are mainly used as induction and maintenance therapy. Moreover, for most of these drugs, a high inter-individual variability in a positive response has been reported [3]. Therefore, it is essential to find clinical efficacious alternatives for the treatment of inflammatory bowel disease (IBD). Recently, intestinal permeability has been recognized as a new target for IBD prevention and therapy [4,5]. In physiological conditions, the intestinal epithelium selectively absorbs nutrients and represents an efficient barrier to noxious antigens and pathogens. Disruption of the intestinal barrier has been considered a major factor in several inflammatory intestinal diseases [6]. The architecture and function of the intestinal epithelium requires close coordination between enterocyte proliferation and apoptosis, both highly dependent on cell–cell and cell–matrix interactions. Enterocytes are joined to each other by tight junctions and adherens junctions. The main molecular component of the adherens junctions is E-cadherin, a transmembrane protein with five extracellular domains that interdigitate with those of an adjacent cell in a calcium-dependent homophilic manner to form a continuous linear “zipper” structure [7,8]. E-cadherin exerts functional adhesion activity when it is connected to the actin cytoskeleton by the cytoplasmic complex α-, β-, or γ-catenin in a mutually exclusive manner [7,8]. It is well established that, in IBD patients, these junctional complexes undergo a disruption leading to a breaking of the intestinal barrier [9]. Moreover, mucosal biopsies from patients with IBD have increased levels of cytokines, able to regulate intercellular permeability, such as interferon-gamma (IFN-γ) and tumor necrosis factor (TNF-α), have been observed [10]. 

Buffalo (*Bubalus bubalis*) milk is a source of many bioactive peptides that have been demonstrated to play a relevant role in preventing various disorders [11,12,13,14]. These bioactive peptides are mainly released during gastrointestinal digestion of milk and milk-products. Recently, novel peptides have been identified from buffalo milk-derived products (MBCP) that are able to reduce H_2_O_2_-induced oxidative stress in intestinal epithelial cells and in erythrocytes [15,16]. Among these, MBCP, a peptide isolated from the buffalo milk after *in vitro* digestion of *Mozzarella di Bufala Campana DOP*, has demonstrated possession of a good stability to brush border exopeptidases and a high bioavailability [16]. In this paper, we evaluated the therapeutic potential of MBCP in IBD. For this purpose, we investigated the effect of MBCP on (i) adjacent junctions conformation and permeability under inflammatory conditions in an intestinal epithelial cell line, Caco-2 cells and (ii) intestinal inflammation and the associated changes in motility in mice.

## 2. Material and Methods

### 2.1. Peptide Synthesis

MBCP was synthetised using the solid-phase method and purified via HPLC after deprotection, as previously described [16]. MBCP was dissolved in water for *in vitro* and *in vivo* experiments. 

### 2.2. Drugs and Reagents

2,4,6-dinitrobenzenesulfonic acid (DNBS), croton oil, fluorescein isothiocyanate (FITC)-conjugated dextran (molecular mass 3–5 kDa), betamethasone, atropine, tubocurarine and neutral red (NR) solution were purchased from Sigma (Milan, Italy). TNF-α was obtained from R&D Systems, Space Import-Export SRL (Milano, Italy). All reagents for cell cultures were obtained from Sigma, Bio-Rad Laboratories (Milan, Italy) and Microtech Srl (Naples, Italy). All chemicals and reagents employed in this study were of analytical grade.

### 2.3. Cell Culture

Caco-2 cells (American Type Culture Collection, Rockville, MD, USA), a human colorectal adenocarcinoma epithelial cell line, were grown at 37 °C in a 5% CO_2_ atmosphere in h-glucose Minimum essential medium MEM containing 1% non-essential amino acids and supplemented with 10% de-complemented fetal bovine serum (FBS), 100 U·mL^−1^ penicillin, 100 mg·mL^−1^ streptomycin, 1% l-glutamine and 1% sodium pyruvate. The growth of the cells was measured by counting the cells with a Coulter counter (Nexcelom, Lawrence, MA, USA, Cellometer Auto1000). All experiments were performed in triplicate. 

### 2.4. Animals

Male adult ICR mice, weighing 20–25 g for upper gastrointestinal transit experiments and 25–30 g for colitis experiments, were purchased from Charles River Laboratories (Calco, Lecco, Italy) and housed in polycarbonate cages under controlled temperature (23 ± 2 °C), constant humidity (60%) and a 12 h light/dark cycle. The animals were acclimatized to their environment at least 1 week under the above reported standard conditions with free access to tap water. Mice were fed *ad libitum* with a standard rodent diet, except for the 24 h immediately before the intracolonic administration of DNBS or the oral administration of the charcoal meal and for 2-h before the oral gavage of croton oil or MBCP. Mice were randomly allocated to different experimental groups and outcome assessments were performed in single-blind. All experiments were approved by the Institutional Animal Ethics Committee for the use of experimental animals and conformed to guidelines for the safe use and care of experimental animals in accordance with the Italian D.L. no. 116 of 27 January 1992 and associated guidelines in the European Communities Council (86/609/ECC and 2010/63/UE).

### 2.5. Cytotoxicity Studies

We evaluated the effect of MBCP on Caco-2 cell viability by using a microplate colorimetric assay that measures the ability of viable cells to incorporate and bind the neutral red (NR), a weak cationic dye, in lysosomes. Cells were plated at the appropriate density to obtain a model of undifferentiated and differentiated cells; then, 5 × 10^3^ and 20 × 10^3^ cells per well in 96-well plates for undifferentiated and differentiated cells, respectively. After 24 hours, cells were exposed to various concentrations of MBCP (9–375 μM) for 48-h. Then, cells were incubated with NR dye solution (50 μg/mL) for 3-h at 37 °C, and finally washed with phosphate buffered saline (PBS) and lysed with 1% acetic acid. The absorbance was read at 540 nm (iMarkTM microplate reader, Bio-Rad, Milano, Italy).

### 2.6. Alkaline Phosphatase (ALP) Activity

ALP activity was used as marker of the degree of differentiation of human Caco-2 cells. Attached and floating cells were washed and lysed with 0.25% sodium deoxycholate. ALP activity was determined using Sigma Diagnostics ALP reagent (no. 245). Total cellular protein content of the samples was determined in a microassay procedure using the Coomassie protein assay reagent kit (Pierce). ALP activity was calculated as nmol/min/mg of protein.

### 2.7. Western Blotting

For cell extract preparation, the cells were washed twice with ice-cold PBS/bovine serum albumin (BSA), scraped and centrifuged for 30 min at 4 °C in 1 mL of lysis buffer (1% Triton, 0.5% sodium deoxycholate, 0.1 M NaCl, 1mM Ethylenediamine tetra-acetic acid (EDTA), pH 7.5, 10 mM Na_2_HPO_4_, pH 7.4, 10 mM Phenylmethyl sulfonyl fluoride, 25 mM benzamidin, 1 mM leupeptin, 0.025 units/mL aprotinin). Colon tissue were lysed in a specific buffer (1% NP-40, 0.1% SDS, 100 mM sodium ortovanadate, 0.5% sodium deoxycholate in RIPA buffer) in the presence of protease inhibitors (4 mg/mL of leupeptin, aprotinin, pepstatin A, chymostatin, PMSF and 5 mg/mL of chymotrypsin-like proteases (TPCK)). The homogenates were sonicated twice by three strokes (20 Hz for 20 s each); after centrifugation for 30 min at 10,000× *g*, the supernatants were stored at 80 °C. Equal amounts of cell and tissues proteins were separated by SDS-PAGE, electrotransferred to nitrocellulose and reacted with the different antibodies. Blots were then developed using enhanced chemoluminescence detection reagents (SuperSignal West Pico, Pierce) and exposed to X-ray film. All films were scanned by using Quantity One software (BioRad laboratories, Hercules, CA, USA).

### 2.8. Immunostaining and Confocal Microscopy

After 24-h and 48-h of incubation with TNF-α (10 µM), TNF-α plus MBCP (18 μM) or TNF-α plus betamethasone (10 µM), Caco-2 cells were fixed for 20 min with a 3% (*w*/*v*) paraformaldehyde solution and permeabilized for 10 min with 0.1% (*w*/*v*) Triton X-100 in PBS at room temperature. To prevent nonspecific interactions of antibodies, cells were treated for 2 h in 5% BSA in PBS. In another set of experiments, Caco-2 cells were treated with the acetylcholine receptor antagonists atropine (10 µM) and tubocurarine (10 µM) for 6-h. Immunostaining was carried out by incubation with anti-E-cadherin, anti-actin and anti-β-catenin antibodies (1:500, Alexa Fluor^®^, BD Pharmingen^TM^). The slides were mounted on microscope slides by Mowiol. The analyses were performed with a Zeiss LSM 510 microscope equipped with a plan-apochromat objective X 63 (NA 1.4) in oil immersion. The nuclei were stained with 4′,6-diamidino-2-phenylindole (DAPI). 

### 2.9. Permeability Assay on Caco-2 Cells

To evaluate cell permeability, Caco-2 cells were seeded into a transwell filters with a pore diameter of 8 μm in 24-well plates at a density of 2.0 × 10^5^ cells/insert. Further cultivation in the same medium as above allowed the cells to spontaneously differentiate and polarize into the epithelial monolayer within 21 days. The basolateral compartment contained 1.5 mL of culture medium while apical compartment contained 0.2 mL. After seeding, the cells were treated with 0.07 µM of MBCP. The culture medium was replaced 3 times/wk. After 21 days, the filters containing the cell monolayers (with or without MBCP) were treated for 2 h with TNF-α (10 μM) and then with a mannitol-lactulose solution (0.05 mmol/L, 0.25 mol/L, 0.2 mL). After 3 hours, the basolateral solution was collected and both mannitol and lactulose concentration were measured by liquid chromatography-mass spectrometry (LC-MS).

### 2.10. Induction of Experimental Colitis

Colitis was induced in the anesthetized mice by the intracolonic administration of 2,4,6-dinitrobenzene sulfonic acid (DNBS) as previously described [17]. Briefly, DNBS (150 mg/kg), dissolved in 50% ethanol (150 µL/mouse), was administrated into the distal colon using a polyethylene catheter (1 mm in diameter) via the rectum (4.5 cm from the anus). Three days after the DNBS administration, all mice were euthanized by asphyxiation with CO_2_, the mice abdomen was opened by a midline incision and the colon removed, isolated from surrounding tissues, length measured, opened along the antimesenteric border, rinsed and weighed and then fixed in 10% formaldehyde for histopathological analysis. MBCP was administrated by oral gavage (10–100 mg/kg) once a day for 3 consecutive days starting 24 h after DNBS administration. Animals were euthanized 2 hours after the last administration of MBCP.

### 2.11. Haematoxylin-Eosin Staining

Paraffin-embedded colon tissues were cut into 5-mm sections. The sections were dewaxed in xylene for 10 min and dehydrated in gradient alcohol. The sections were then stained with haematoxylin for 8 min and eosin for 5 min. After dehydration, the sections were sealed and examined under a light microscope (Leica DM 2500). Photographs were taken using the Leica DFC320 R2 digital camera [18].

### 2.12. Immuno-Fluorescence Microscopy

The fixed slides of colon tissue, were dewaxed, rehydrated and processed. Briefly, antigen retrieval was performed by pressure-cooking slides for 3 min in 0.01 M citrate buffer (pH 6.0). To prevent nonspecific interactions of antibodies, the slides were treated for 2 h in 5% BSA in PBS. Immunostaining was carried out by incubation, overnight at 4 °C, with anti-E-cadherin and anti-β-Catenin antibodies (1:100, Alexa Fluor^®^, BD Pharmingen^TM^). The slides were mounted on microscope slides by Mowiol + DAPI for nuclear staining, and then observed under the optical microscope (Leica DM 5000 B + CTR 5000).

### 2.13. Intestinal Permeability Measurement

The effect of MBCP, at a dose of 100 mg/kg, was also tested on intestinal permeability, using the FITC-Dextran method.Briefly, 2 days after the induction of colitis, mice were gavaged with 600 mg/kg of FITC–conjugated dextran (molecular mass 3–5 kDa). After 24 h, blood was collected by cardiac puncture, and the FITC-derived fluorescence was immediately analyzed in the serum by a microplate reader (GloMax Explorer System, Promega; excitation wavelengths 485 ± 14 nm, emission wavelengths 520 ± 25 nm). Serial-diluted FITC dextran was used to generate a standard curve. Intestinal permeability was expressed as the concentrations of FITC (μM) detected in the serum.

### 2.14. Induction of Intestinal Hypermotility and Upper Gastrointestinal Transit in Mice

Upper gastrointestinal transit was measured in both physiological and pathological conditions. The hypermotility was induced by the inflammatory agent croton oil (CO) as previously described [19]. Briefly, two doses of CO (20 µL/mouse) were given for two consecutive days by oral gavage and the upper gastrointestinal transit was measured 4 days after the first administration of CO [20] (i.e., when the maximal inflammatory response associated to the intestinal hypermotility was reported). Mice were deprived of food overnight and then the upper gastrointestinal transit was evaluated by identifying the leading front of an intragastrically administered charcoal meal marker (10% charcoal suspension in 5% gum Arabic, 10 mL/kg) in the small intestine as previously described [19]. Twenty minutes after charcoal administration, mice were euthanized and the small intestine was isolated by cutting at the pyloric and ileocaecal junctions. The distance traveled by the marker was measured and expressed as a percentage of the total length of the small intestine from pylorus to caecum. MBCP was administrated by oral gavage (5–50 mg/kg) 30 min prior to charcoal administration.

### 2.15. Statistical Analysis

Data are expressed as the mean ± S.E.M. or S.D. of *n* experiments. Statistical significance was assessed using the Student’s *t*-test for comparing a single treatment mean with a control mean, and a one-way ANOVA followed by a Tukey multiple comparisons test for the analysis of multiple treatment means. Values of *p* < 0.05 were considered significant. The IC_50_ (concentration that produced 50% inhibition of cell viability) values were calculated using sigmoidal dose response curve-fitting models (Graphpad Prism Software, version 5.03, Inc. Avendia de la Playa La Jolla, CA, USA).

## 3. Results

### 3.1. In Vitro Study

#### 3.1.1. MBCP Induces Cytotoxic Effects in Caco-2 Cells at Very High Concentrations

In order to characterize a safety profile of MBCP, we evaluated its effect on the cell viability of pre- (undifferentiated exponentially growing) and post-confluent (differentiated) Caco-2 cells. MBCP (9–375 μM), induced in undifferentiated and differentiated Caco-2 cells, a concentration-dependent cell viability inhibition at 48 h with an IC_50_ of 65 μM (Figure 1). Therefore, to characterize the anti-inflammatory effects of MBCP on Caco-2 cells, subsequent experiments were performed using a MBCP concentration that did not induce significant cytotoxic effect in both pre and post-confluent cells (i.e., 18 μM). 

#### 3.1.2. MBCP Stimulates Differentiation in Growing Caco-2 Cells

The effect of MBCP on cell differentiation was evaluated by measuring the activity of alkaline phosphatase, with the enzyme frequently used as a marker of colon cells differentiation [21]. A treatment of undifferentiated Caco-2 cells with MBCP (18 μM for 48 h) induced a significantly increase (*p* < 0.05) of alkaline phosphatase activity (APA) of about 14% compared to the untreated Caco-2 cells (control). The APA value was in the Caco-2 cells of 8.6 ± 0.4 (nmol/min/mg protein), while, in MBCP treated Caco-2 cells, it was 10 ± 0.7 (mean ± SD, *n* = 3, * *p* < 0.05) 

#### 3.1.3. MBCP Modulates Adherent Junctions Formation in Control and TNF-α-Stimulated Caco-2 Cells

In order to investigate the role of MBCP on adherent junctions (AJs) formation, we studied the cellular organization of E-cadherin and actin in pre-confluent Caco-2 cells by confocal miscroscopy. Caco-2 cells, after 24 h from seeding, showed a strong cytoplasmic localization of both E-cadherin and actin proteins (Figure 2A–C) that was ameliorated by MBCP treatment (Figure 2D–F). At 48 h, the E-cadherin-actin complex presented a not homogeneous membranous localization in untreated Caco-2 cells (Figure 2G–I). MBCP treatment induced a strong cell–cell contact membranous localization of E-cadherin-Actin complex (Figure 2M–O). 

To assess whether MBCP could counteract the AJ-destructuration induced by the pro-inflammatory mediator TNF-α, we examined the β-catenin organization in TNF-α-stimulated cells. Caco-2 cells treated with TNF-α (10 µM, for 48-h) showed a cytoplasmic compartment β-catenin localization (Figure 3D–F) compared to untreated cells (CTR) (Figure 3A–C). A treatment of Caco-2 cell with MBCP (18 µM) induced an enhancement of both E-cadherin and β-catenin localization at AJs (Figure 2G–I) compared to TNF-α alone. Since corticosteroids are the effective first-line treatment in cytokine-induced inflammation, we treated the Caco-2 cells (stimulated with TNF-α) with betamethasone. Betamethasone (10 μM) treatment induced a membranous E-cadherin and β-catenin organization, although a β-catenin cytoplasmic immunoreactivity persisted (Figure 3L–N). In our experimental conditions (Caco-2 cells treated or not with TNF-α), MBCP accelerated and induced an organization of adjacent junctions in the cells, without changes the E-cadherin, actin and β-catenin protein expression evaluated by Western blot analysis (data not shown). 

#### 3.1.4. MBCP Counteracts the TNF-α Inflammatory Effect by Modulating the NF-κB Pathway

It is well known that TNF-α causes the activation of transcription factors, including NF-κB. NF-κB regulates host inflammatory, immune responses and also stimulates the expression of inducible cyclooxygenase enzymes (COX-2) that contribute to the pathogenesis of the inflammatory process. Therefore, we evaluated, by Western blot analysis, the effects of MBCP on NF-κB, pNF-κB, COX-2 and 5-Lipoxygenases (LOX) expression in TNF-α treated Caco-2 cells. A treatment with TNF-α induced an increase of 5-LOX, p-NF-κB and COX-2 expression. MBCP (18 µM) was able to reduce the expression of p-NF-κB, 5-LOX and COX-2 expression increased by TNF-α (Figure 4). Moreover, MBCP (18 µM) reduced the basal expression of NF-κB in untreated Caco-2 cells. Betamethasone (10 µM), used as a positive control, significantly reduced 5-LOX expression, but not the NF-κB, p-NF-κB and COX-2 expression, increased by TNF-α.

#### 3.1.5. MBCP Reduces Atropine and Tubocurarine-Induced Adherens Junctions Disorganization

Several studies reported that the stimulation of the acetylcholine receptors (AChR) enhances epithelial barrier formation [22,23] and ameliorates TNF-α induced barrier dysfunction in intestinal epithelial cells [24]. According to literature, treatment of Caco-2 cells with the AChR antagonists atropine (10 µM) or tubocurarine (10 µM) induced an disorganization of adherents junctions (Figure 5). MBCP (18 µM) attenuated both atropine and tubocurarine AChR antagonist effects on Caco-2 cells adherens junctions, indicating that MBCP treatment could affect the AChR pathway (Figure 5). 

#### 3.1.6. MBCP Inhibits the TNF-α-Increased Caco-2 Permeability

The intestinal epithelial cell line Caco-2 has been used extensively as a model of the human epithelium, as it can be grown in the Transwell system as a differentiated cell monolayer that has selective paracellular permeability to ions and solutes. We used both mannitol and lactulose, intestinal permeability probes, to evaluate the permeability changes in Caco-2 stimulated with TNF-α. Treatment of the Caco-2 cell (see Section 2.9) with TNF-α (10 μM) increased the concentration of mannitol in the basolateral solution by about twofold compared to the control cells (cells without TNF-α treatment). [Control of mannitol concentration (mmol/L): 0.01 ± 0.004; TNF-α 0.023 ± 0.005 *; mean ± SD; * *p* < 0.05]. The mannitol concentration in the basolateral side of the Caco-2 cells pretreated with MBCP and then incubated with TNF-α (0.009 ± 0.003 mmol/L, mean ± SD; n = 3, * *p* < 0.05) was decreased twofold compared to the TNF-α treated Caco-2 cells. The MBCP induced any change of the lactulose permeability in the same inflammatory cell condition. 

### 3.2. In Vivo Study

#### 3.2.1. MBCP Reduces the Inflammation in the DNBS Model of Colitis

The DNBS murine model of colitis was used to assess the in vivo intestinal anti-inflammatory effects of MBCP. DNBS administration (150 mg/kg) caused inflammatory damage, as indicated by the approximately twofold increase in colon weight/colon length *ratio* (mg/cm), a simple and reliable marker of inflammation and damage (Figure 6A). MBCP (10–100 mg/kg, by oral gavage), administered for three consecutive days after the inflammatory insult, significantly and in a dose-dependent manner, reduced the effect of DNBS on colon weight/colon length *ratio.* The effect was significant starting from 30 mg/kg dose (Figure 6A). The anti-inflammatory effect MBCP was further confirmed by histological analysis. As shown in Figure 6B, DNBS caused a severe inflammatory cellular infiltration and complete destruction of the colon epithelium, compared to control mice (Figure 6B). Our data showed that oral MBCP (100 mg/kg) reduced the colonic damage induced by DNBS (Figure 6B). 

Considering that the peptide inhibited the phosphorylation of NF-κB in vitro, we investigated if this pathway was involved in the in vivo anti-inflammatory effect of MBCP. As shown in Figure 6C, the phosphorylation of IκBα and active NF-κB expression were increased in the colons of mice treated with DNBS. MBCP (100 mg/kg) was able to significantly reduce these changes induced by DNBS. These results demonstrated that MBCP suppressed NF-κB activation both in the experimental model of colitis and in TNFα-stimulated Caco-2 cells.

#### 3.2.2. MBCP Reduces the Intestinal Permeability in Vivo

The intestinal permeability was increased of the intracolonic administration of DNBS, as revealed by the high concentration of FITC-conjugated dextran in the serum (see Appendix A). While MBCP (100 mg/kg), given by oral gavage for three consecutive days, synergistically (*p* < 0.01) partially counteracted the DNBS-induced increase in intestinal permeability (see Appendix A). Moreover, immunofluorescence analysis showed that DNBS administration caused a destructuration of the colonic AJs associated with an increase of the citoplasmatic expression of E-cadherin and β-catenin. MBCP (100 mg/kg) counteracted the effect of DNBS on AJs, (Figure 7), thus confirming the *in vitro* results on TNF-α-stimulated Caco-2 cells.

#### 3.2.3. MBCP Counteracts the Accelerated Upper Gastrointestinal Transit Induced by Croton Oil

The administration of the flogogen agent croton oil (CO) induced an accelerated upper gastrointestinal transit (GT) 4 days after its first administration. The physiological GT percentage in the mice (control) was of 49.8 ± 1.4; whereas GT significantly increased up to values of 65.50 ± 6.26* (mean ± SEM; * *p* < 0.05) in CO treated mice. MBCP (5–50 mg/kg), given by oral gavage 30 min before the administration of the charcoal, in a dose dependent manner restored the intestinal motility to physiological conditions (% of GT MBCP 5 mg/kg: 59.43 ± 2.75; MBCP 10 mg/kg: 56.14 ± 3.63; MBCP 50 mg/kg: 39.87 ± 6.38 **; ** *p* < 0.01). MBCP, at the high dose (50 mg/kg), did not modify the upper gastrointestinal transit in control mice (% of GT: control 55 ± 3.53; MBCP 48.60 ± 4.38 (mean ± SEM)).

## 4. Discussion

In mammals, the enterocytes are renewed continuously every 4–8 days through an organized series of events involving proliferation, differentiation and programmed cell death. The proliferation to differentiation transition (PDT) is a critical step in the continual renewal of a normal intestinal epithelium [25]. A useful *in vitro* model to study the effects of food ingredients on the gut epithelial layer is based on the use of Caco-2 cells for their functional similarity to colonic enterocytes [26]. Indeed, Caco-2 cells express tight junctions, microvilli, enzymes and transporters functionally similar to colonic enterocyte [26]. Moreover, this cell line has the ability to elicit a pro-inflammatory reaction in response to stimulants like TNF-α, a known mediator of gastrointestinal mucosal barrier injury [5,27]. In this study, we have shown that a peptide (MBCP) obtained from gastrointestinal digestion of *Mozzarella of Bufala Campana DOP* is able to modulate the differentiation and permeability in Caco-2 cells stimulated with TNF-α and to attenuate inflammation and hypermotility in murine models of intestinal inflammation. 

### 4.1. MBPC Modulates the Differentiation and Permeability in Caco-2 Cells

Proliferating Caco-2 cells spontaneously initiate the differentiation process when they have reached confluence. The differentiation program starts when specific biochemical events induce the cell–cell contact, through the E-Cadherin/actin/*β*-Catenin complex [28]. Here, we have found that MBCP, at a non-cytotoxic concentration (i.e., 18 µM), induced an increase of intestinal alkaline phosphatase activity in undifferentiated Caco2 cells. Intestinal alkaline phosphatase is a well-known cell differentiation marker and its activity has been inversely correlated with an increased risk of intestinal inflammation development [29]. Moreover, we also found that MBCP treatment, already after 24 h, induced the differentiation program accelerating the E-cadherin-actin membranous organization, thus suggesting a beneficial effect on the AJ. This result is further supported by the experiments with TNF-α. According to previous studies [30], we have shown that Caco-2 treatment with TNF-α induced molecular alterations of the E-cadherin organization and consequently a passage of mannitol through the Caco-2 monolayer. MBCP treatment was able to restore the cell–cell junctions, counteracting the breakdown of E-cadherin-β-catenin complex and reducing the increase of mannitol permeability inducted by TNF-α. 

According to previous studies, we found that un-stimulated Caco-2 cells over-expressed COX-2 and NF-κB and TNF-α-stimulated Caco-2 cells increased 5-LOX expression. NF-κB transcription factor is a master regulator of the inflammatory response and it is essential for the homeostasis of the immune system. NF-κB regulates the transcription of genes, such as LOX and COX-2, which control inflammation [31]. Our results demonstrated that MBCP reduced the phosphorylation of NF-κB as well as COX-2 and 5-LOX expression in TNF-α stimulated cells. 

Many mechanisms are responsible for the regulation and stability of adherens junction proteins, one of these possibilities could be the AChR activation. Furthermore, previous studies indicate that cholinergic agonists interfere with NF-κB pathways preventing IkBα breakdown and p65 nuclear translocation [24,32]. Moreover, cholinergic agonists could prevent gut barrier failure after severe burn injury maintaining intestinal barrier integrity. McGilligan has showed that nicotine decrease Caco-2 permeability by regulating the expression of TJ proteins [33]. Therefore, we measured changes in localization of β-catenin to determine whether modulation of this protein correlates with the adherens junctions disorganization induced by cholinergic antagonists like atropine and tubocurarine. Confocal microscopy analysis confirmed that both atropine and tubocurarine treatment induced a cytoplasmic accumulation of β-catenin in Caco-2 cells. These detrimental effects induced by cholinergic antagonists were counteracted by MBCP treatment.

### 4.2. MBPC Ameliorates Murine Colitis

The ability of MBCP to restore cell–cell contacts as well as to exert an anti-inflammatory actions was subsequently evaluated *in vivo* by using the mice model of DNBS-induced colitis. According to previous studies [34,35] intracolonic administration of DNBS induced intestinal inflammation associated to an increase of epithelial permeability. Oral MBCP administration was able to reduce intestinal inflammation as demonstrated by the reduction of colon weight colon length *ratio* (a simple and reliable marker of inflammation and damage), histological alterations, IκBα phosphorylation and of NF-κB activation associated with DNBS administration. 

Intestinal permeability plays a crucial role in the development of IBD as well as in the IBD ongoing bowel symptoms [5,36]. Moreover, inflammation reduces barrier integrity and affects the normal intestinal permeability [4]. Studies on Caco-2 cells have shown that MBCP restored tight junctions altered by TNF-α. Tight junctions are multi-protein complexes that maintain the intestinal barrier while regulating permeability [9]. Therefore, we investigated the effect of MBPC on intestinal permeability in vivo. We adopted an immunofluorescent method through which an orally-administered marker (i.e., FITC-conjugated dextran) can be detected in the blood if permeability is impaired. As expected, intestinal permeability increased after DNBS administration and, more importantly, MBCP restored the impaired permeability.

### 4.3. MBPC Normalizes Inflammation-Induced Murine Intestinal Hypermotility

It is clinically well established that inflammation in the gut causes debilitating symptoms due to motility disturbances [37]. To investigate the effect of MBCP on intestinal motility, we administered mice croton oil, which is a flogogen agent able to induce hypermotility as a consequence of intestinal inflammation. By using this experimental model, we have shown that MBPC did not affect motility in healthy mice, but normalized the exaggerated intestinal transit caused by the inflammatory insult. The lack of effect of MBPC in control mice is clinically relevant in the light of the observation that constipation is a very common side effect associated with drugs clinically used to reduce intestinal motility [38,39].

## 5. Conclusions

In conclusion, the data obtained in our study shown that MBCP, a peptide isolated from the buffalo milk after *in vitro* digestion of *Mozzarella di Bufala Campana DOP*, exerts anti-inflammatory effects both *in vitro* and *in vivo*. This anti-inflammatory effect could be related to its beneficial effects on adherens junctions mainly during an inflammatory process. These results could have an important impact on the therapeutic potential of MBCP in helping to restore the intestinal epithelium integrity damaged by inflammation, thereby reducing the risk of colorectal cancer.

## Figures and Tables

**Figure 1 nutrients-11-00610-f001:**
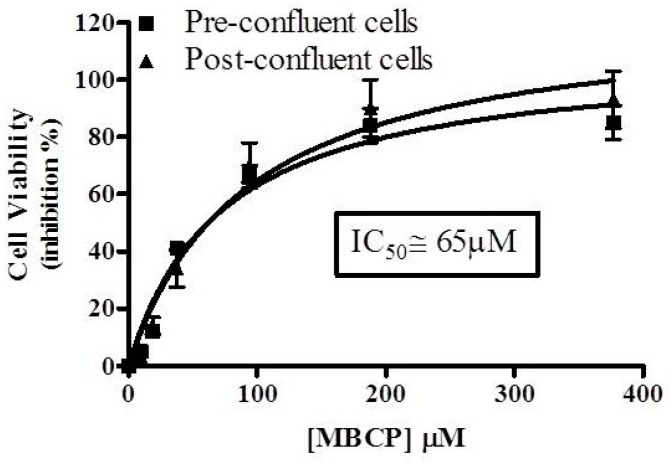
MBCP reduces cell viability in Caco-2 cells. Cell viability was evaluated by the Neutral Red assay in pre and post-confluent Caco-2 cells. Cells were incubated with increasing concentration of MBCP (9–375 µM) for 48 h. Each point represents the mean ± SD of three independent experiments.

**Figure 2 nutrients-11-00610-f002:**
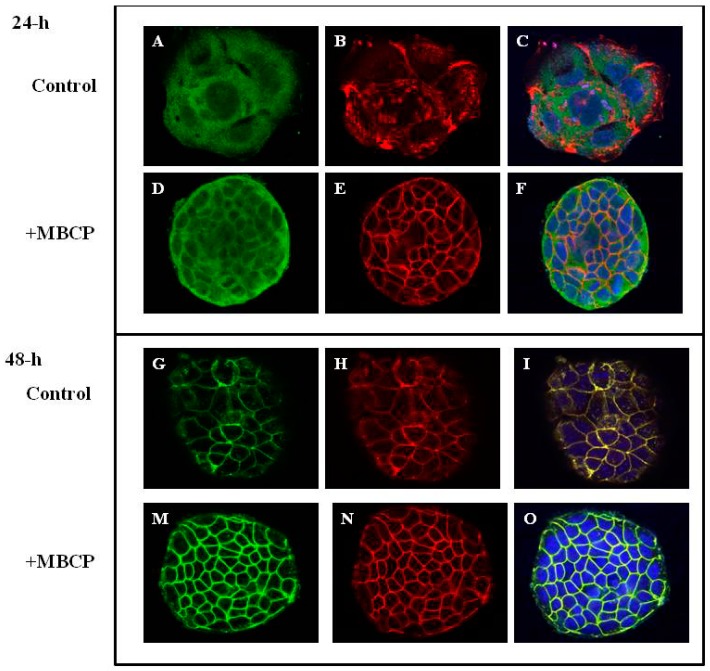
MBCP accelerates the formation of adherens junctions. Confocal microscopy images of Caco-2 cells (**A**–**C**,**G**–**I**) and MBCP treated Caco-2 cells (panel (**D**–**F**,**M**–**O**). Cells were treated for 24-h and 48-h with MBCP at 18 μM concentration. The merged image is on the right (green, actin; red, E-cadherin; blue, dapi).

**Figure 3 nutrients-11-00610-f003:**
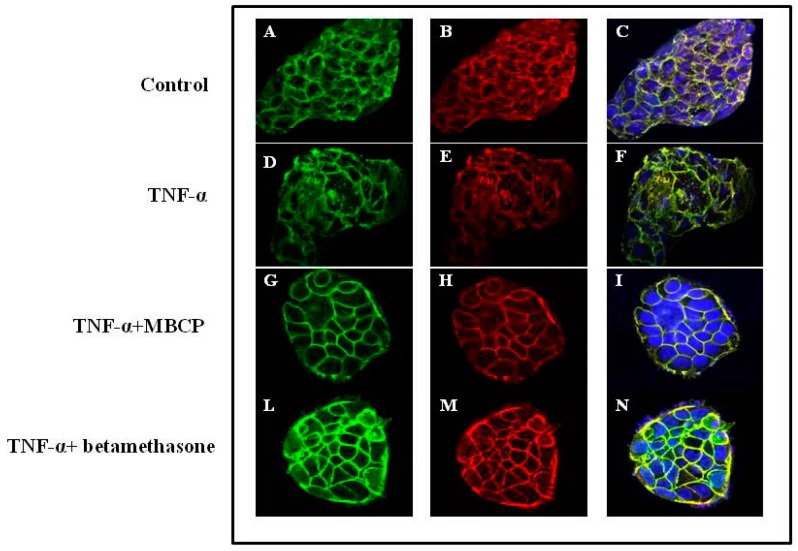
MBCP ameliorates the adherens junctions destructured by TNF-α. Confocal microscopy images of untreated Caco-2 cells (panel (**A**–**C**), Caco-2 cells treated for 48 h with TNF-α (10 μM, (**D**–**F)**), TNFα 10μM plus MBCP (18 µM, (**G**–**I)**) or with TNF-α (10 μM) plus betamethasone (**L**–**N**). On the right, merged image (green, β-catenin; red, E-cadherin; blue, dapi).

**Figure 4 nutrients-11-00610-f004:**
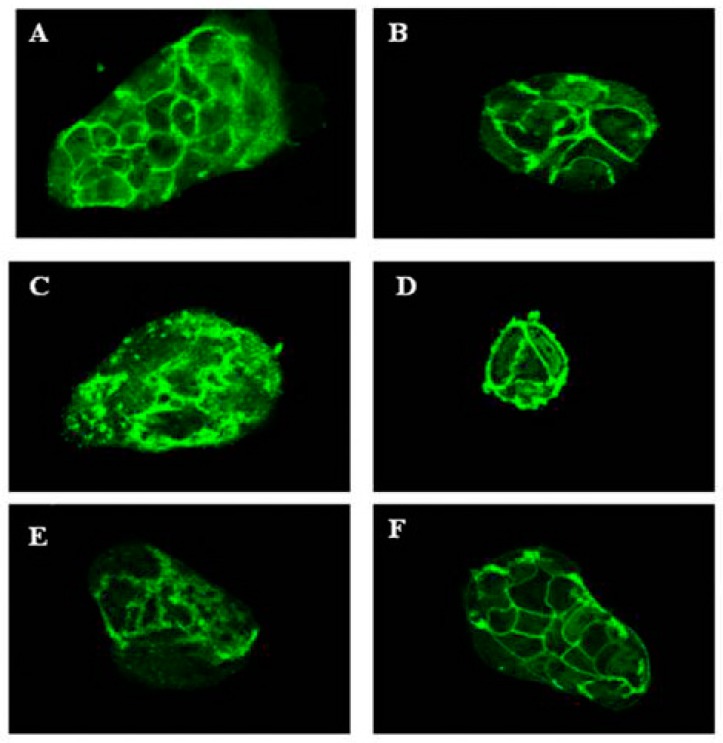
MBCP ameliorates the adherens junctions destructured by atropine and tubocurarine. Confocal microscopy images of untreated Caco-2 cells (**A**) or Caco-2 cells treated for six hours with MBCP (18 µM, (**B**)), tubocurarine (10 µM, (**C**)), Tubocurarine plus MBCP (**D**), atropine (10 µM, (**E**)) and atropine plus MBCP (**F**). The color green represents β-catenin.

**Figure 5 nutrients-11-00610-f005:**
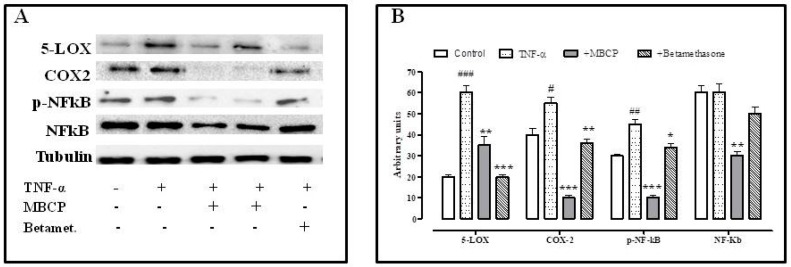
MBCP reduces the expression of 5-LOX, COX-2, p-NF-κB and NF-κB. (**A**) Western blot analysis of Caco-2 cells alone or treated with MBCP or betamethasone. Bands associated with the expression of 5-LOX, COX-2, p-NF-κB, NF-κB, and house-keeping γ-tubulin, after 48 h of treatment with MBCP (18 µM) and betamethasone (10 µM). The expression of the house-keeping protein γ-tubulin was used as loading control. (**B**) The expression levels of the above proteins are reported as a percentage with respect to the level of the protein in untreated cells, used as control. Bars show the mean ± SD of three different determinations. # *p* < 0.05, ## *p* < 0.01 and ### *p* < 0.001 vs. control; * *p* < 0.05, ** *p* < 0.01 and *** *p* < 0.001 vs. TNF-α alone.

**Figure 6 nutrients-11-00610-f006:**
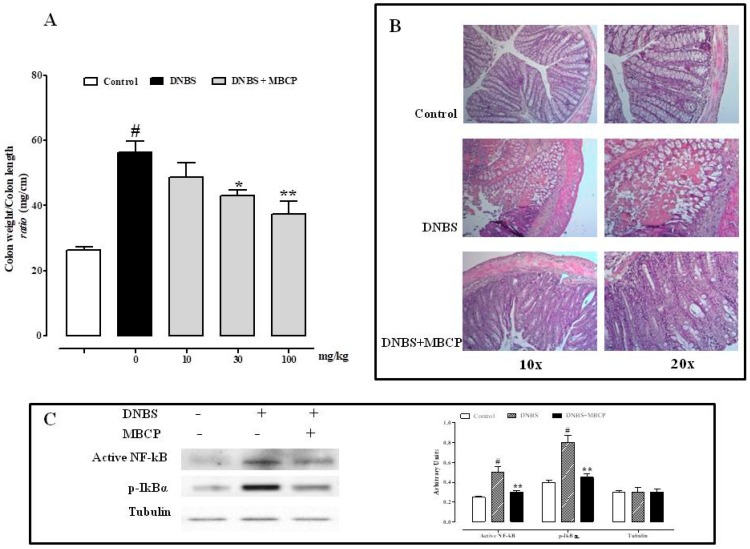
MBCP reduces colon weight/colon length *ratio* in DNBS-induced colitis in mice. (**A**) MBCP (10–100 mg/kg, by oral gavage) was administered once a day starting 24 h after the induction of colitis by DNBS (150 mg/kg). Colons were collected three days after DNBS. (**B**) representative hematoxylin & eosin stained colon cross-sections of mice treated with vehicle (control), DNBS and DNBS plus MBCP (100 mg/kg by oral gavage). Colons were collected three days after the induction of colitis by DNBS. Original magnification 10× and 20× (A). (**C**) Western blot analysis of p-NF-κB, NF-κB, and house-keeping γ-tubulin expression in normal mice colon and after the induction of colitis by DNBS.All data are represented as mean ± SEM of seven mice for each experimental group. Statistical significance was calculated using one-way ANOVA test. # *p* < 0.001 vs. control, * *p* < 0.05 and ** *p* < 0.01 vs. DNBS alone.

**Figure 7 nutrients-11-00610-f007:**
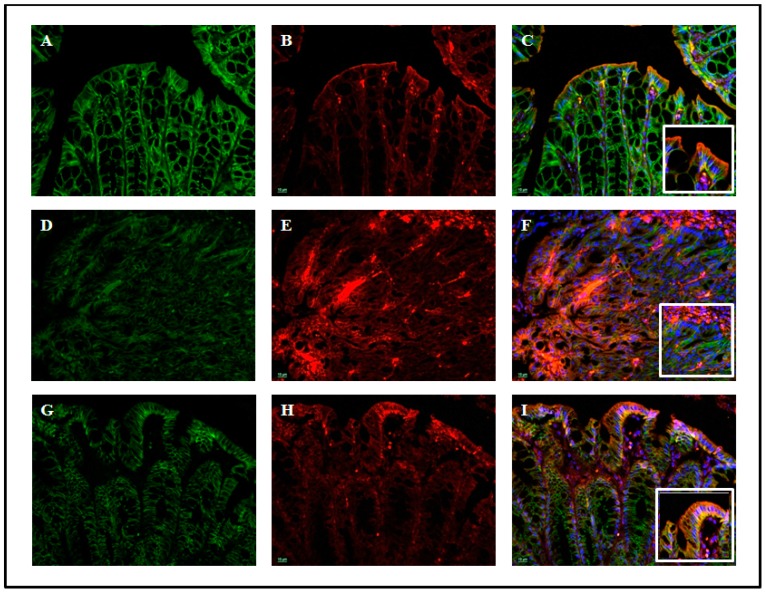
MBCP induces the organization of colonic AJs in DNBS-induced colitis in mice. Immunofluorescence analysis showing the expression of E-cadherin (red) and β-catenin (green) in control (**A**–**C**), DNBS (**D**–**F**) and DNBS plus MBCP (**G**–**I**) (100 mg/kg) mice. Original magnification 20×. A magnified portion of the immunofluorescence was shown at higher magnification in the inset of panel (**C**,**F**,**I**). The DNBS-induced colitis in mice was treated for three consecutive days after the inflammatory insults with MBCP (100 mg/kg, by oral gavage).

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
