# Peer review of "Intestinal Anti-Inflammatory Effect of a Peptide Derived from Gastrointestinal Digestion of Buffalo (Bubalus bubalis) Mozzarella Cheese"

_nutrients, 2019, doi:10.3390/nu11030610_

Reviewer 1 Report

The manuscript deals with the analysis the effect of MBCP on adjacents junctions conformation and permeability under inflammatory conditions in an intestinal epithelial cell line, Caco-2 cells and intestinal inflammation and the associated changes in motility in mice. In general, the work is interesting and fits well with the Journal's scope. The methodology used for these analyses is adequate, although there are some aspects that could be improved.

For example, certain fragments of the manuscript should be described more clearly.

Another issue is that the some of the figures presented in the dimly visible way.

Specific comments:

If we use name abbreviations, they should be explained on the first use (Line 28, and check in all manuscript).

Line 34 and 436: or to be sure in vitro and in vitro is correct?

Line 116: there is no bracket behind respectively.

Line 130-135: different font.

Line 157: avoid writing shortcuts.

Line 234-236 and 301-306 and 353-360 This paragraphs should be more clearly described.

Lines 241: there should be Figure 2 and not Figure 1.

Figure 5 B is illegible.

All these aspects should be clarified in the text.

Author Response

Point by Point Response to Reviewers

Article code:nutrient-449049.

Title: Intestinal anti-inflammatory effect of a peptide derived from gastrointestinal digestion of buffalo (Bubalus bubalis) mozzarella cheese

Authors: Giancarlo Tenore, Ester Pagano, Stefania Lama, Daniela Vanacore, Salvatore di Maro, Maria Maisto, Raffaele Capasso, Francesco Merlino, Francesca Borrelli, Paola Stiuso, Ettore Novellino.

We thank the reviewers for their comments. In the manuscript all revisions were made with the "Track Revisions" function of Microsoft Word.

Answer Reviewer #1

Comment 1 of the reviewer: If we use name abbreviations, they should be explained on the first use (Line 28, and check in all manuscript).

Answer: The abbreviations were explained in all manuscript.

Comment 2 of the reviewer:Line 34 and 436: or to be sure in vitro and in vitro is correct?

Answer: We have corrected at line 34 and 436  the sentence " in vitro and in vitro " with " in vitro and in vivo"

Comment 3 of the reviewer: Line 116: there is no bracket behind respectively.

Answer: The sentence: "Cells were plated at the appropriate density to obtain a model of differentiated and undifferentiated cells (5 × 103 and 20 × 103 cells per well in 96-well plates for undifferentiated and differentiated cells, respectively)." were changed with: "Cells were plated at the appropriate density to obtain a model of undifferentiated and differentiated cells,  then 5 × 103 and 20 × 103 cells per well in 96-well plates for undifferentiated and differentiated cells, respectively."

Comment 4 of the reviewer: Line 130-135: different font.

Answer:We have correct the font in line130-135.

Comment 5 of the reviewer: Line 157: avoid writing shortcuts.

Answer.The sentence " The basolateral (serosal) and apical (mucosal) compartments contained 1.5 and 0.2 mL of culture medium, respectively" was correct with " The basolateral compartment contained 1.5 mL of culture medium while  apical compartment contained  0.2 mL."

Comment 6 of the reviewer:line 234-236 this paragraphs should be more clearly described.

Answer: The paragraph :"The effect of MBCP on cell differentiation was evaluated by measuring the activity of alkaline phosphatase which is an enzyme frequently used as a marker of cell differentiation in colon cells (21) . A treatment of pre-confluent Caco-2 cells with MBCP (18 μM for 48-h) induced an alkaline phosphatase activity increase of about 14% compared to the untreated Caco-2 cells [alkaline phosphatase activity (nmol/min/mg protein) Control: 8.6±0.4; MBCP (18 μM) 10 ±0.7*; mean±SD, n=3*, p<0.05]" has been replaced with:" The effect of MBCP on cell differentiation was evaluated by measuring the activity of alkaline phosphatase, enzyme frequently used as a marker of colon cells differentiation (21). A treatment of undifferentiated Caco-2 cells with MBCP (18 μM for 48-h) induced a significantly increase (p<0.05) of alkaline phosphatase activity (APA)  of about 14% compared to the untreated Caco-2 cells (control). The APA value was,  in the Caco-2 cells of 8.6±0.4 (nmol/min/mg protein); while in MBCP   treated Caco-2 cells was 10 ±0.7 ( mean±SD, n=3) .

Comment 7 of the reviewer:line 301-306 this paragraphs should be more clearly described.

Answer: The paragraph: "We used mannitol for assessing the permeability changes in TNFα-stimulated Caco-2 treated with  MBCP. The treatment of Caco-2 cell with TNF-α (10 µM) increased the mannitol concentration in the basolateral solution [mannitol concentration (mmol/L) control: 0.01±0.004; TNF-α 0.023±0.005*; mean±SD; *p<0.05]. MBCP (0.07 µM) decreased of two fold the mannitol concentration augmented by TNF-α [mannitol concentration (mmol/L) TNF-α 0.02±0.005; TNF-α plus MBCP: 0.009± 0.003*; mean±SD; n=3, *p<0.05]"< p="">

 has been replaced with: " We used both mannitol and lactulose, intestinal permeability probes, to evaluate the permeability changes in Caco-2 stimulated with TNF-α. Treatment of the Caco-2 cell (see section 2.9) with TNF-α (10 μM) increased the concentration of mannitol in the basolateral solution by about 2-fold compared to the control cells (cells without TNF-α treatment). [Control of mannitol concentration (mmol / L): 0.01 ± 0.004; TNF-α 0.023 ± 0.005 *; mean ± SD; * P<0.05]. The mannitol concentration in the basolateral side of the Caco-2 cells pretreated with MBCP and than incubated with TNF-α  (0.009 ± 0.003 mmol / L, mean ± SD; n = 3, * p <0.05).was decreased of 2 fold compared to the TNF-α treated Caco-2 cells. The MBCP induced any change of the lactulose permeability in same inflammatory cell condition.

Comment 8 of the reviewer:line 353-360 this paragraphs should be more clearly described.

Answer: The paragraph: "The administration of the flogogen agent croton oil (CO) induced an accelerated upper gastrointestinal transit 4 days after its first administration [% of transit (mean±SEM): control 49.8±1.4; CO 65.50±6.26*; n=3, *p<0.05]. MBCP (5-50 mg/kg), given by oral gavage 30 min before the administration of the charcoal, in a dose dependent manner restored the intestinal motility to physiological conditions [% of transit (mean±SEM): CO 65.5±6.26; MBCP 5 mg/kg: 59.43± 2.75; MBCP 10 mg/kg: 56.14±3.63; MBCP 50 mg/kg: 39.87±6.38**; **p<0.01]. MBCP, at the high dose (50 mg/kg), did not modify the upper gastrointestinal transit in control mice [% of transit (mean±SEM): control 55±3.53; MBCP 48.60±4.38].

has been replaced with: "The intestinal permeability was increased of the intracolonic administration of DNBS, as revealed by the high concentration of FITC-conjugated dextran in the serum (see Supplementary files). While MBCP (100 mg/kg), given by oral gavage for three consecutive days, synergistically (p<0.01) partially counteracted the DNBS-induced increase in intestinal permeability (see Supplementary files). Moreover, immunofluorescence analysis showed that DNBS administration caused a destructuration of the colonic AJs associated to an increase of the citoplasmatic expression of E-cadherin and β-catenin. MBCP (100 mg/kg) counteracted the effect of DNBS on AJs, (Figure 7) thus confirming the in vitro results on TNF-α-stimulated Caco-2 cells."

Comment 9 of the reviewer:Lines 241: there should be Figure 2 and not Figure 1.

Answer: we have changed the figure 2 with figure 1

Comment 10 of the reviewer: Figure 5 B is illegible.

Answer: we have changed the figure 5.

Reviewer 2 Report

The authors have done a good job but there are few fundamental questions to be answered:

1. the authors have used Caco 2 cell lines and shown increased permeability but in those experiments the authors did not show decrease in E-cadherin or beta catenin.  the authors should show decrease in the said adherent junctions and then with MBCP there is increase in the said adherent junctions would definitely proof that these adherent junctions proteins are involved. the authors should do western blot and show the decrease and then increase with MBCP.
2. Similarly in in vivo studies also the authors have not shown increase or decrease of adherent junctions.  The authors have shown in the figure 7 the merged picture of the E-cadherin and B-catenin but the authors should show individual pics and then merged pic.  the authors have said that E-cadherin and B-catenin show citoplasmatic expression but with the pics it is not all possible to decide and hence the authors should show quantitative data with the fluorescent pics. But the best method would be showing the changes in protein expression using western blot.
3.The intestinal permeability is caused by upregulation of claudin 2.  There are quite few published manuscript like Tissue Barriers. 2015; 3(1-2): e977176.  the authors in their whole manuscript did not mention claudin 2 any  time.  any reason and in both in vitro as well as in vivo the authors never checked for the claudins?
4.  The authors show both in vivo as well as in vitro the MBCP blocks NFkB but did the blockage help decrease TNFalpha or Il6 in serum or tissue homogenate would be good.

Author Response

Point by Point Response to Reviewers

Article code:nutrient-449049.

Title: Intestinal anti-inflammatory effect of a peptide derived from gastrointestinal digestion of buffalo (Bubalus bubalis) mozzarella cheese

Authors: Giancarlo Tenore, Ester Pagano, Stefania Lama, Daniela Vanacore, Salvatore di Maro, Maria Maisto, Raffaele Capasso, Francesco Merlino, Francesca Borrelli, Paola Stiuso, Ettore Novellino.

We thank the reviewers for their comments. In the manuscript all revisions were made with the "Track Revisions" function of Microsoft Word.

Answer Reviewer #2

Comment 1 of the reviewer: the authors have used Caco 2 cell lines and shown increased permeability but in those experiments the authors did not show decrease in E-cadherin or beta catenin.  the authors should show decrease in the said adherent junctions and then with MBCP there is increase in the said adherent junctions would definitely proof that these adherent junctions proteins are involved. the authors should do western blot and show the decrease and then increase with MBCP.

Answer: at the Line 254 we have add the sentence: " In our experimental conditions (with and without TNF-α treated Caco-2 cells) MBCP accelerated and induced  an organization of adjacent junctions in the cells, without changes the E-cadherin, actin and  β-catenin proteins expression evaluated by western blot analysis (Data not shown).

Comment 2 of the reviewer: Similarly in vivo studies also the authors have not shown increase or decrease of adherent junctions.  The authors have shown in the figure 7 the merged picture of the E-cadherin and b-catenin but the authors should show individual pics and then merged pic.  the authors have said that E-cadherin and B-catenin show citoplasmatic expression but with the pics it is not all possible to decide and hence the authors should show quantitative data with the fluorescent pics. But the best method would be showing the changes in protein expression using western blot.

Answer: we have modified the figure 7 and the paragraph "3.2.2. MBCP reduces the intestinal permeability in vivo"

Comment 3 of the reviewer:The intestinal permeability is caused by upregulation of claudin 2.  There are quite few published manuscript like Tissue Barriers. 2015; 3(1-2): e977176.  the authors in their whole manuscript did not mention claudin 2 any  time.  any reason and in both in vitro as well as in vivo the authors never checked for the claudins?

Answer: Claudin-2 is upregulated in the small and large intestine during intestinal inflammation (Crohn's disease, ulcerative colitis, ibd) and contributes to diarrhea via a leak flux mechanism. In parallel increase the discontinuities of tight junction strand. In our experimental condition we determined the in vitro permeability of mannitol and lactulose in Caco-2 cells treated with TNF-alpha. We have observed that the MBCP induced a change only in the mannitol permeability. Because the mannitol enters the cell trought the hydrophilic portion of the cell membrane is not influenced by the TJ, we have not dealt with the protein components of the TJ.

The paragraph " The intestinal epithelial cell line Caco-2 has been used extensively as a model of the human epithelium, as it can be grown in the Transwell system as a differentiated cell monolayer that has selective paracellular permeability to ions and solutes. We used mannitol for assessing the permeability changes in TNFα-stimulated Caco-2 treated with  MBCP. The treatment of Caco-2 cell with TNF-α (10 µM) increased the mannitol concentration in the basolateral solution [mannitol concentration (mmol/L) control: 0.01±0.004; TNF-α 0.023±0.005*; mean±SD; *p<0.05]. MBCP (0.07 µM) decreased of two fold the mannitol concentration augmented by TNF-α [mannitol concentration (mmol/L) TNF-α 0.02±0.005; TNF-α plus MBCP: 0.009± 0.003*; mean±SD; n=3, *p<0.05]. "

 has been replaced with: " We used both mannitol and lactulose, intestinal permeability probes, to evaluate the permeability changes in Caco-2 stimulated with TNF-α. Treatment of the Caco-2 cell (see section 2.9) with TNF-α (10 μM) increased the concentration of mannitol in the basolateral solution by about 2-fold compared to the control cells (cells without TNF-α treatment). [Control of mannitol concentration (mmol / L): 0.01 ± 0.004; TNF-α 0.023 ± 0.005 *; mean ± SD; * P<0.05]. The mannitol concentration in the basolateral side of the Caco-2 cells pretreated with MBCP and than incubated with TNF-α  (0.009 ± 0.003 mmol / L, mean ± SD; n = 3, * p <0.05).was decreased of 2 fold compared to the TNF-α treated Caco-2 cells. The MBCP induced any change of the lactulose permeability in same inflammatory cell condition.

Comment 4 of the reviewer: The authors show both in vivo as well as in vitro the MBCP blocks NFkB but did the blockage help decrease TNF-alpha or Il6 in serum or tissue homogenate would be good.

Answer: The MBCP treatment in DNBS mice model not shown significantly variation of TNF-alpha or Il6 concentration.

Round  2

Reviewer 1 Report

In my opinion that work is okay.

Author Response

 Point by Point Response to Reviewers
Article code:nutrient-449049.
Title: Intestinal anti-inflammatory effect of a peptide derived from gastrointestinal digestion of buffalo (Bubalus bubalis) mozzarella cheese

Authors: Giancarlo Tenore, Ester Pagano, Stefania Lama, Daniela Vanacore, Salvatore di Maro, Maria Maisto, Raffaele Capasso, Francesco Merlino, Francesca Borrelli, Paola Stiuso, Ettore Novellino.

We thank the reviewer .

Reviewer 2 Report

The authors done given answer to most of the queries.  There are few minor defects in the manuscript like:

  In the material and methods section 2. 7 Western Blotting line 135 supernatant were stored at 80 oc. It should be minus 80oC.

the authors have misspelled cytoplasmic compartment as citoplasmic.  Please change them.

Would love to see a more magnified version of the Immunostaining of E-cadherin n B-catenin in both Caco-2 and in vivo studies.Just higher magnification of the photos.

Author Response

Point by Point Response to Reviewers

Article code:nutrient-449049.

Title: Intestinal anti-inflammatory effect of a peptide derived from gastrointestinal digestion of buffalo (Bubalus bubalis) mozzarella cheese

Authors: Giancarlo Tenore, Ester Pagano, Stefania Lama, Daniela Vanacore, Salvatore di Maro, Maria Maisto, Raffaele Capasso, Francesco Merlino, Francesca Borrelli, Paola Stiuso, Ettore Novellino.

We thank the reviewer for their comments. In the manuscript all revisions were made with the "Track Revisions" function of Microsoft Word.

Reviewer #2

The authors done given answer to most of the queries.  There are few minor defects in the manuscript like:

Comment 1 of the reviewer: In the material and methods section 2. 7 Western Blotting line 135 supernatant were stored at 80 oc. It should be minus 80oC.

Response 1: we have modified, in the material and methods section 2. 7 Western Blotting line 135, the font of the sentence.

Comment 2 of the reviewer: the authors have misspelled cytoplasmic compartment as citoplasmic.  Please change them.

Response 2: we have corrected the citoplasmic as cytoplasmic compartment.

Comment 3 of the reviewer: Would love to see a more magnified version of the Immunostaining of E-cadherin n B-catenin in both Caco-2 and in vivo studies. Just higher magnification of the photos.

Response 3: we have modified the figure 7 with a more magnified version.
